# Long-Term Persistence of Three Microbial Wildfire Biomarkers in Forest Soils

Antonio J. Fernández-González [1,*], Ana V. Lasa [1], José F. Cobo-Díaz [2], Pablo J. Villadas [1],
Antonio J. Pérez-Luque [3], Fernando M. García-Rodríguez [1], Susannah G. Tringe [4]
and Manuel Fernández-López [1,*]

[1]  Department of Soil and Plant Microbiology, Estación Experimental del Zaidín, CSIC (Consejo Superior de Investigaciones Cinetíficas), Profesor Albareda 1, 18008 Granada, Spain; ana.vicente@eez.csic.es (A.V.L.); pablo.villadas@eez.csic.es (P.J.V.); fernando.garcia@eez.csic.es (F.M.G.-R.)

[2]  Department of Food Hygiene and Technology, Faculty of Veterinary, Universidad de León, c\ Campus de Vegazana s/n, 24007 León, Spain; jcobd@unileon.es

[3]  Department of Assessment, Restoration and Protection of Mediterranean Agrosystems (SERPAM), Estación Experimental del Zaidín, CSIC, Profesor Albareda 1, 18008 Granada, Spain; antonio.perez@eez.csic.es

[4]  DOE Joint Genome Institute, Lawrence Berkeley National Laboratory, 1 Cyclotron Road, Berkeley, CA 94720, USA; sgtringe@lbl.gov

*  Correspondence: antonio.fernandez@eez.csic.es (A.J.F.-G.); manuel.fernandez@eez.csic.es (M.F.-L.)

**Abstract:** Long-term monitoring of microbial communities in the rhizosphere of post-fire forests is currently one of the key knowledge gaps. Knowing the time scale of the effects is indispensable to aiding post-fire recovery in vulnerable woodlands, including holm oak forests, that are subjected to a Mediterranean climate, as is the case with forests that are found in protected areas such as the Sierra Nevada National and Natural Park in southeastern Spain. We took rhizosphere soil samples from burned and unburned holm oak trees approximately 3, 6, and 9 years after the 2005 fire that devastated almost 3500 ha in southeastern Spain. We observed that the prokaryotic communities are recovering but have not yet reached the conditions observed in the unburned forest. A common denominator between this fire and other fires is the long-term persistence of three ecosystem recovery biomarkers—specifically, higher proportions of the genera *Arthrobacter*, *Blastococcus*, and *Massilia* in soil microbial communities after a forest fire. These pyrophilous microbes possess remarkable resilience against adverse conditions, exhibiting traits such as xerotolerance, nitrogen mineralization, degradation of aromatic compounds, and copiotrophy in favorable conditions. Furthermore, these biomarkers thrive in alkaline environments, which persist over the long term following forest fires. The relative abundance of these biomarkers showed a decreasing trend over time, but they were still far from the values of the control condition. In conclusion, a decade does not seem to be enough for the complete recovery of the prokaryotic communities in this Mediterranean ecosystem.

**Keywords:** forest fire; prokaryotic community; rhizosphere; *Arthrobacter*; *Blastococcus*; *Massilia*

## 1. Introduction

Holm oak forests, consisting mainly of various *Quercus ilex* L. subspecies, are among the natural vegetation in zones that are subjected to a Mediterranean climate. The Mediterranean climate occurs in a transition zone between the temperate forests and the shrublands that predominate in tropical zones. Its most outstanding characteristics are cold winters and hot summers with prolonged droughts. In addition, holm oak forests are distributed in both natural environments, such as the Spanish *dehesas* (forested rangelands), and areas with a high level of human activity, such as largely human-populated and tourist areas, which span over 3 million ha in the Spanish territory [1]. However, the present vegetation coverage has decreased significantly compared to the original native forest, predominantly as a result of human activities [2]. In addition, the abandonment of rural

areas raises concerns about the possible loss of management needed in these ecosystems. Another factor to consider, according to a vegetation dynamics model trained with data from southern Spain and France, is the loss of holm oak forests due to their conversion to shrublands, mainly because of the increased water stress suffered in these areas as a result of global warming and fires [3]. Therefore, all the above-mentioned factors make this type of vegetation particularly sensitive to stresses such as drought and wildfires, exacerbated by climate change.

In particular, forest fires in Spain have increased in recent decades from an average of 1871 fire events in the 1960s to an average of 13,111 fire events in the decade from 2006 to 2015. In addition, in 2010s, the affected forest area in all cases exceeded 50,000 ha per year, with an average of 113,848 ha per decade, reaching 188,697 ha in the most catastrophic year (2005) [4]. Among the 25,492 fire events recorded that year in Spain, one occurred in the Sierra Nevada National and Natural Park and affected almost 3500 ha, of which more than 400 ha were dominated by holm oak trees [5,6].

Soil microbial communities play a pivotal role in the biogeochemical processes within ecosystems, such as carbon (C) and nitrogen (N) cycling. Such processes are responsible for decomposing organic matter, releasing nutrients, and facilitating the nutrient uptake by plants [1,3,6]. When wildfires occur, they can significantly alter the composition and activities of soil microbial communities. The severity of a fire, such as its intensity and duration, can directly influence the extent of these changes. Therefore, studying the connections between wildfire and community characteristics provides us with a foundation for comprehending the consequences of fire on ecosystem dynamics. [7,8].

There has been an increase not only in the number of fires, but also in the interest in studying their effects on the microbial communities that inhabit the disturbed vegetation, both in the short term (up to 1 year after a fire) [7–12] and in the intermediate term (around 3 to 6 years after a fire) [13–16]. To the best of our knowledge, when a fire occurs, it induces a reduction in diversity and an alteration in the composition of microbial communities, as many of them are unable to withstand this disturbance in the short term and the intermediate term [10,11]. Such changes are evident in terms of both the composition and the functionality of the community, with an increased abundance of copiotrophic microorganisms that intensifies with the fire severity [15]. Furthermore, during these post-fire periods, an enrichment in the metagenome of individuals exhibiting pyrophilic traits is observed. In other words, as previously mentioned, these microorganisms possess accelerated growth rates as well as enhanced heat tolerance and the capability to process pyrogenic compounds [12]. Consequently, in the short-to-intermediate term, the microbial community displays greater resilience to post-fire conditions.

However, studies focusing on the long-term monitoring of microbial community recovery during the process of ecosystem restoration are not common. Moreover, such studies on the rhizosphere microbiome of burned trees are scarce [14,17], in spite of the known role of microorganisms in promoting plant growth and resilience [18]. In this sense, our previous studies showed the key role of the bacterium *Arthrobacter* in fire restoration. This genus was not only the most abundant in the rhizosphere of the burned holm oak forest, with a relative abundance above 20%, but it was also an essential agent in plant growth promotion [14], with the potential to create a nitrogen-rich environment [6]. Therefore, *Arthrobacter* is a soil microbial biomarker of fire that has been detected in environments as different as that of our previous studies, in a holm oak forest subjected to a Mediterranean climate, and a boreal forest with a mix of conifers in Canada [10]. How biomarkers such as *Arthrobacter* change over the long term is poorly explored, as the few studies that followed such microbial communities for many years (i.e., close to or exceeding a decade) were not conducted longitudinally, i.e., by following the evolution of the microbial communities of the same fire and the same individual trees [19–23].

Given the limited number of long-term monitoring studies conducted on post-fire forests, and considering that the existing studies are reliant on pre-high-throughput sequencing methods, it is imperative to undertake research employing contemporary tech-

niques. Such studies will enable us to acquire a more comprehensive understanding of the modifications experienced by rhizosphere microbial communities inhabiting these environments. In the current study, we present, for the first time, a longitudinal analysis in the intermediate term (3 and 6 years after a fire) and the long term (9 years after a fire) of the rhizosphere prokaryotic communities of the holm oak forest after the above-mentioned fire occurred in 2005, with the main objective of understanding the resilience of such a threatened and sensitive ecosystem. To achieve this objective, we tried to answer the following questions: (i) Were both the diversity and structure of the prokaryotic community and the physicochemical parameters of the soil re-established almost a decade after fire? (ii) How did the fire affect the prokaryotic biomass? (iii) Are there any genera that act as key indicators in the recovery process of these communities? (iv) Could pyrogenic carbon have an effect on the medium-to-long term maintenance of the alteration in the prokaryotic profile?

## 2. Materials and Methods

### 2.1. Experimental Site

The study area is situated within the Sierra Nevada Natural and National Park (SE Spain). It encompasses an area where a wildfire occurred in September 2005, resulting in the burning of 3427 ha, including 412 ha of centenary evergreen holm oaks (*Quercus ilex* subsp. *ballota*). The valley of the Lanjarón river was selected as the collection site for rhizosphere samples. Two specific locations were chosen: one within a holm oak forest that was severely impacted by the wildfire (referred to as the burned oak forest, BOF; N 36°57′26″, W 3°27′48″; 1566 m above sea level [m.a.s.l.]) and the other in an adjacent undisturbed holm oak forest that was unaffected by the fire (known as the undisturbed oak forest, UOF; N 36°58′11″, W 3°27′37″; 1790 m.a.s.l.). The undisturbed oak forest has not experienced any recorded fires in the past century (see Supplementary Materials: Figure S1). The BOF and the UOF are the same sites that were previously described [6,14] and the sampled trees and the first sampling time (3 years after the fire) were also the same. A longitudinal sampling was followed at 3, 6, and 9 years after the fire (yaf) to study the recovery of the rhizosphere prokaryotic communities from the holm oaks in the medium term and the long term. The two sampling sites were located on a steep slope facing south. Within each study site, three sampling plots were randomly selected along 1.0 km transects. The rhizosphere of three trees was collected per plot (2 sites × 3 plots/site × 3 trees/plot = 18 trees); each tree had a diameter of at least 15 cm at breast height and was separated by at least 5 m from the other trees. The uppermost layer of soil (first 5 cm) was carefully removed and, subsequently, rhizosphere soil samples were obtained from a depth ranging between 5 and 20 cm. The collection process involved tracing the primary roots of each plant until reaching non-suberified roots, where the soil adhered to the roots was manually gathered.

To facilitate monitoring and longitudinal tracking, the sampled trees were marked with green paint spray and their locations were recorded using a Garmin GPS model GPSmap60CSx (Olathe, KS, USA). Soil samples were promptly preserved in 50 mL Falcon tubes at −80 °C, and within the subsequent 24 h and prior to freezing, 0.25 g of each sample were subjected to DNA extraction procedures. Furthermore, soil under the canopy of each tree was collected and pooled from each plot [n = 3 per condition (burn status) and year] then sieved through a 2 mm mesh for physicochemical analysis, at the first time-point (3 yaf) and the last time-point (9 yaf). These analyses (including soil type, pH, available water, electrical conductivity, organic matter, total nitrogen, carbon-to-nitrogen ratio, available Pi, and K) were performed with standardized procedures at the Food and Agriculture Laboratory of the Andalusian regional government at Atarfe (Granada, Spain).

### 2.2. DNA Extraction, PCR Amplification, and Pyrosequencing

Soil DNA was isolated from each individual sample (*n* = 9 per condition and time-point) using the PowerSoil™ DNA Isolation Kit (MoBio, Laboratories Inc., Carlsbad, CA,

USA), following the recommended protocols provided by the manufacturer. The extracted DNA was evaluated for both quantity and quality. Quantification was performed using a Nanodrop ND-1000 spectrophotometer (Nanodrop Technologies, Wilmington, DE, USA), while quality assessment was conducted by analyzing the DNA samples through electrophoresis in a 0.8% (*w/v*) agarose gel stained with Gel Red under UV light.

To prepare for pyrosequencing, equal amounts of DNA from the three trees within the same plot were pooled together (*n* = 3 per condition and time-point). Amplification of prokaryotic hypervariable V6-V8 *16S rRNA* gene regions was carried out using the previously described primers: 926F (5′-AAACTYAAAKGAATTGRCGG-3′) and 1392R (5′-ACGGGCGGTGTGTRC-3′) [14,24]. Subsequently, all amplicons were combined in equimolar proportions, resulting in a total of 18 composite samples (*n* = 2 conditions × 3 plots × 3 time-points) that were subjected to pyrosequencing with the Genome Sequencer Titanium FLX system (454 | Life Sciences, Branford, CT, USA) at the Joint Genome Institute (Walnut Creek, CA) in the case of the first 2 sampling times and with the Genome Sequencer Junior (454 | Life Sciences, Branford, CT, USA) at Estación Experimental del Zaidín—CSIC (Granada, Spain) for the last sampling time.

### 2.3. Pyrosequencing Data Analysis

The raw sequences obtained were processed using MOTHUR version 1.39.5 [25,26]. To minimize sequencing errors, the AmpliconNoise algorithm was employed, and low-quality sequences were eliminated based on specific criteria, including a minimum length of 150 bp, allowance for 2 mismatches in the barcode and primer sequences, and exclusion of homopolymers exceeding 8 bp. Subsequently, sequence alignment was performed using *align.seqs* function and the SILVA database (NR version 123) as template [27]. The *chimera.uchime* function was applied to identify and remove potentially chimeric sequences [28]. Following these steps, the remaining high-quality prokaryotic sequences were clustered into operational taxonomic units (OTUs) at a similarity threshold of 97%. Finally, OTUs were classified by matching them against the Ribosomal Database Project (RDP-II) 16S rRNA reference database, training set v.16 MOTHUR-formatted, by using an 80% bootstrap cutoff for classification.

### 2.4. Statistical Analyses from NGS Data

The number of sequences per sample was rarefied only before performing alpha diversity analysis [29]. Split-plot design ANOVA (SPANOVA) or two-way mixed-design ANOVA was performed to compare alpha diversity indices (observed richness, Shannon, the inverse of Simpson, and Pielou's evenness) between conditions (control vs. burned) and among time-points (3, 6, and 9 yaf) using the *anova_test* function of R package *rstatix* [30] and holm as the false discovery rate (FDR) multiple-test correction. Furthermore, generalized eta squared (ges) was assessed to obtain the effect size of statistically significant differences with the same function. Previously, normality and homoscedasticity were assessed with Shapiro (*aov* function from *stats*) and Levene (*leveneTest* function from *car*) tests, respectively. For beta diversity analysis, the filtered OTU sequence counts were normalized using the "trimmed means of M" (TMM) method, implemented with the BioConductor package *edgeR* [31]. The normalized data were used to conduct a principal coordinates analysis (PCoA) on weighted UniFrac distances. This PCoA analysis allowed for the two-dimensional ordination of beta diversity variance between different conditions and across time-points. The ordination analysis was executed using the R package *phyloseq* [32].

To assess the significance of the observed differences, a permutational analysis of variance (PERMANOVA) and a permutational analysis of multivariate dispersions (BETADISPER) were performed. These analyses were conducted using the *adonis* and *betadisper* functions in the Vegan package, employing 9999 permutations [33]. Furthermore, a constrained analysis of principal coordinates (CAP) was conducted based on weighted UniFrac distances. This analysis aimed to examine the impact of physicochemical soil properties on microbial profiles. In summary, the independent parameters were selected using the

*capscale* function, considering only those with variance inflation factors (VIF) below 10. Subsequently, ANOVA tests using the *anova.cca* function were performed to identify statistically significant parameters in the resulting distribution. Moreover, with the function *cor.test*, the correlations between the significantly different parameters and the relative abundance of bacterial genera were computed. Those significant correlations ($p < 0.05$) with Spearman *rho* $\geq$ absolute value of 0.6 were considered strong correlations. Finally, differentially abundant phyla and genera were obtained with function *ancomloop*, an implementation of the *ANCOM-BC* package that is publicly available in our microbial amplicon analysis workflow [34].

### 2.5. Prokaryotic and Actinobacterial Biomass Quantification

Real-time quantitative PCRs (qPCRs) were performed to measure the abundance of total prokaryotic and actinobacterial *16S rRNA* genes. The amplifications were optimized with a gradient of annealing temperature (from 52 to 63 °C) with template DNA from sample BOF 3 years after the fire. PCR reactions were performed as described by Cébron et al. [35], using an Eppendorf 5331 MasterCyclerGradient Thermal Cycler. Amplified fragments were visualized and compared with ø29 HindIII digested DNA (4370 to 72 bp) marker on 1% $w/w$ agarose gels stained with GelRed.

qPCRs of the two populations were carried out in the iCycler iQ system (Bio-Rad Laboratories, Inc., Hercules, CA, USA) using the primer pair E27F (5′-AGAGTTTGATCMTGGCT CAG-3′) and U341R (5′-CTGCTGCSYCCCGTAGG-3′) for the total prokaryotic community, which worked successfully in Medina et al. [36]. For actinobacterial population, 3AF (5′-TATCAGGAGGAACACCGA-3′) and Act2R (5′-AGCCTTGGTAAGGTTCTTCG-3′) primer pair was designed and the specificity to phylum *Actinobacteria* checked in silico using Geneious R6 and Primer-BLAST and in vitro via PCR and electrophoresis in agarose gel (see Supplementary Materials: Figure S2). Actinobacterial qPCR conditions were the same as that of the whole prokaryotic community. Dilution series from $10^8$ to $10^1$ copies of pGEM-T with *16rRNA* gene from *E. coli* were used as control to perform the standard curves for quantitative analysis. Values of threshold cycles (Ct) were determined and the target gene copy number in the samples was calculated from standard curves. qPCR conditions, PCR inhibitor, and the purity of amplified products were performed as described by [35–37]. The results were expressed as copy number per g dry soil and used as a proxy for prokaryotic and actinobacterial biomass. A two-samples *t*-test using the *rstatix* package in R with Holm's method to adjust the *p*-value was performed for comparisons among the conditions and the collection time-points.

### 2.6. Pyrogenic Carbon Assessment

Rhizosphere soils of each plot ($n$ = 3 per condition and time-point), stored at −80 °C, were thawed and weighted 15 g for polycyclic aromatic hydrocarbon (PAH) measurement. Theses GC-MS analyses were carried out at the Instrumental Technical Services of the Estación Experimental del Zaidín—CSIC (Granada, Spain) with a Varian 450-GC attached to a 240-IT mass spectrometer. The two-samples *t*-test using the *rstatix* package in R with Holm's method to adjust the *p*-value was performed for comparisons among the conditions and the collection time-points.

### 2.7. Satellite Monitoring of Fire

Fire severity was evaluated using the delta normalized burn ratio (dNBR) [38], which is the most frequent metric used in the literature to estimate burn severity from satellite data [39]. The dNBR is defined as:

$$dNBR = \left( NBR_{prefire} - NBR_{postfire} \right) \times 1000 \tag{1}$$

and the normalized burn ratio (NBR) is defined as:

$$\text{NBR} = \left( \frac{\rho_{\text{NIR}} - \rho_{\text{SWIR}}}{\rho_{\text{NIR}} + \rho_{\text{SWIR}}} \right) \tag{2}$$

where $\rho_{\text{NIR}}$ and $\rho_{\text{SWIR}}$ are the reflectance of the near infrared and shortwave infrared bands, respectively. This satellite-inferred fire severity metric was produced using Landsat Enhanced Thematic mapper Plus (ETM+) images of our study area, with a pixel resolution of 30 m. We used Google Earth Engine [40] to compute the mean pre- and post-fire NBR values across a date range (01 June to 31 October, 2005) for each valid pixel (i.e., cloud and snow-free pixels). For this purpose, the Landsat Surface Reflectance Tier 1 datasets were used, which include a quality assessment mask (CFMask, see [41]) to identify those pixels with clouds, shadows, water, and snow. Once invalid pixels were excluded, mean composites of pre- and post-fire NBR were produced, and the dNBR was computed. This mean-compositing approach renders the need for a priori scene selection unnecessary [42]. Each pixel was then categorized according to its dNBR value using the classification proposed by the European Forest Fire Information Service (EFFIS) [43] (Table 1).

**Table 1.** Differenced normalized burn ratio (dNBR) thresholds proposed by the European Forest Fire Information Service (EFFIS).

| EFFIS Thresholds | Severity Level |
|:---:|:---:|
| dNBR < 0.100 | Unburned/Very Low |
| $0.100 \le$ dNBR $\le 0.255$ | Low |
| $0.256 \le$ dNBR $\le 0.419$ | Moderate |
| $0.420 \le$ dNBR $\le 0.660$ | High |
| dNBR > 0.660 | Very High |

*2.8. Post-Fire Vegetation Recovery*

To assess the post-fire vegetation recovery, we used the normalized difference vegetation index (NDVI). We used Landsat products (Landsat 5, 7 and 8) with a 30-m spatial resolution to create a 22-year time series (from 2000 to 2021) of annual NDVI for each of the pixels covering the sampling plots. The NDVI data used in this study were derived using the Landsat-based Detection of Trends in Disturbance and Recovery algorithm (LandTrendr) [44] through its Google Earth Engine implementation [45]. Furthermore, we compared the pre-fire NDVI values (2002–2005; reference) with NDVI annual values of 3, 6, and 9 years after fire of the sampling plots. For this purpose, we used non-parametric Wilcoxon tests.

*2.9. Climatic Data*

To characterize the climatic context, we used meteorological data from the Arquilla 6257I station (Network of Secondary Stations of the Spanish Meteorological Agency), which is located at 1652 m.a.s.l. in the municipality of Lanjarón. Data were downloaded using Climanevada [46] and a rainfall temporal series from 1990 to 2017 was generated. Then, we computed the cumulative rainfall for each hydrological year (from 01 October). Additionally, the cumulative rainfall in the 120 days prior to the sampling dates was computed.

**3. Results**

*3.1. Temporal Recovery in Microbial Diversity 9 Years after Fire*

Regarding $\alpha$-diversity, similar values were shown in all indices at the final sampling time (9 yaf) when comparing the burned samples against control (Figure 1a). Unexpectedly, no differences were found between forest status (unburned versus burned; SPANOVA test $p = 0.425$), according to observed richness values. However, both conditions (unburned and burned) showed statistically significantly lower numbers of OTUs at the initial sampling time (3 yaf) when compared to the other two time-points (*t*-test pairwise comparisons

$p < 0.01$). Notwithstanding the result shown by the observed richness, differences were observed in the diversity indices (Shannon and the inverse of Simpson) and in the evenness index when comparing forest status, sampling time, and the interaction of both factors (SPANOVA tests $p < 0.01$; ges > 0.4). Indeed, differences among sampling times were only shown in burned forest samples (Figure 1a) although they were statistically significant only in the evenness index (*t*-test pairwise comparisons $p < 0.05$) and marginally significant in both diversity indices (*t*-test pairwise comparisons $p < 0.1$).

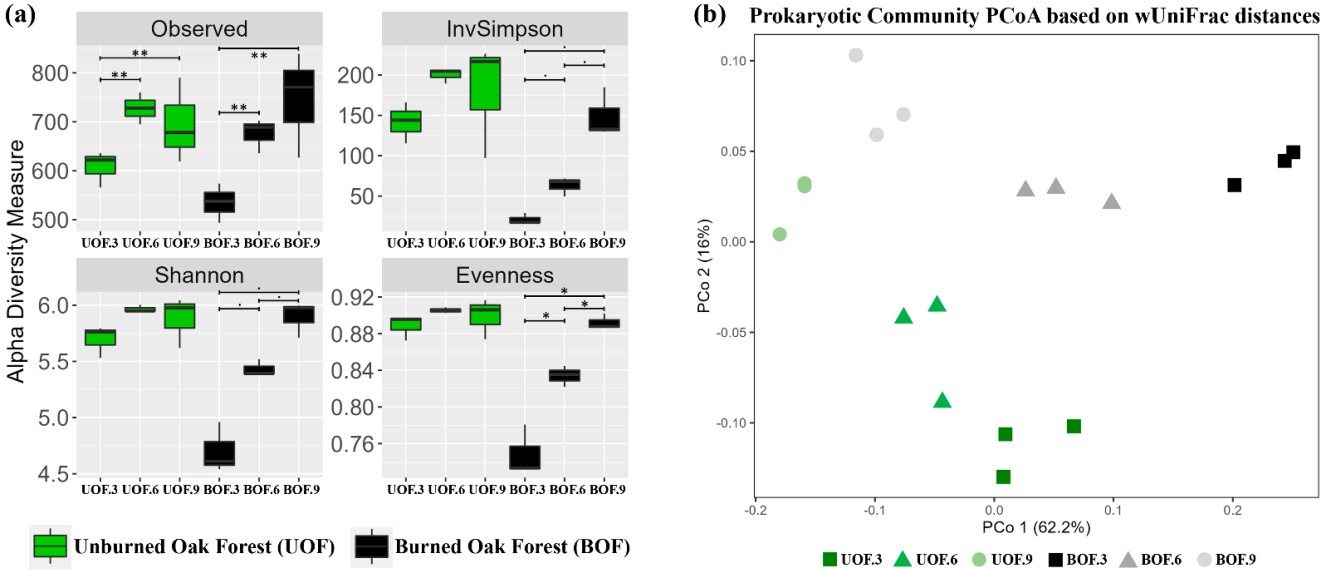

**Figure 1.** Alpha (**a**) and beta (**b**) diversity of prokaryotic communities. In panel a, *t*-test pairwise comparisons are shown as a double asterisk when $p < 0.01$; as a single asterisk when $p < 0.05$; and as a dot when $p < 0.1$.

In terms of beta-diversity, the most remarkable result was that the prokaryotic community observed at 9 yaf was significantly different from those sampled 3 and 6 yaf (PERMANOVA test $p < 0.001$, $R^2 = 0.50$; BETADISPER test $p = 0.297$), regardless of forest status (Figure 1b). In addition, when prokaryotic communities of unburned and burned forest were compared, statistically significant differences were obtained (PERMANOVA test $p < 0.001$; $R^2 = 0.25$; BETADISPER test $p > 0.315$). For instance, considering the distribution of samples in the final situation (9 yaf), we can see a clear separation (Figure 1b), although smaller than in previous years, between the communities coming from each condition. As expected, the biggest separation between burned and unburned samples was observed at the initial sampling point (3 yaf). Finally, significant interaction between both factors was detected (PERMANOVA test $p = 0.01$; $R^2 = 0.09$), which justified their separation in subsequent section plots.

### 3.2. Taxonomic Profiles Remained Markedly Different from Each Other

The holm oak rhizosphere prokaryotic community from Sierra Nevada Natural and National Park was composed of 17 different phyla and, on average, the 10 most abundant accounted for 90% of the high-quality sequences (Figure 2a). On the one hand, an after-fire status comparison five of the main phyla displayed statistically significant differences. For instance, *Actinobacteria* and *Firmicutes* increased in the burned forest (BOF) with respect to control (UOF) and *Proteobacteria*, *Acidobacteria*, and *Verrucomicrobia* decreased with the fire effect (Figure 2a).

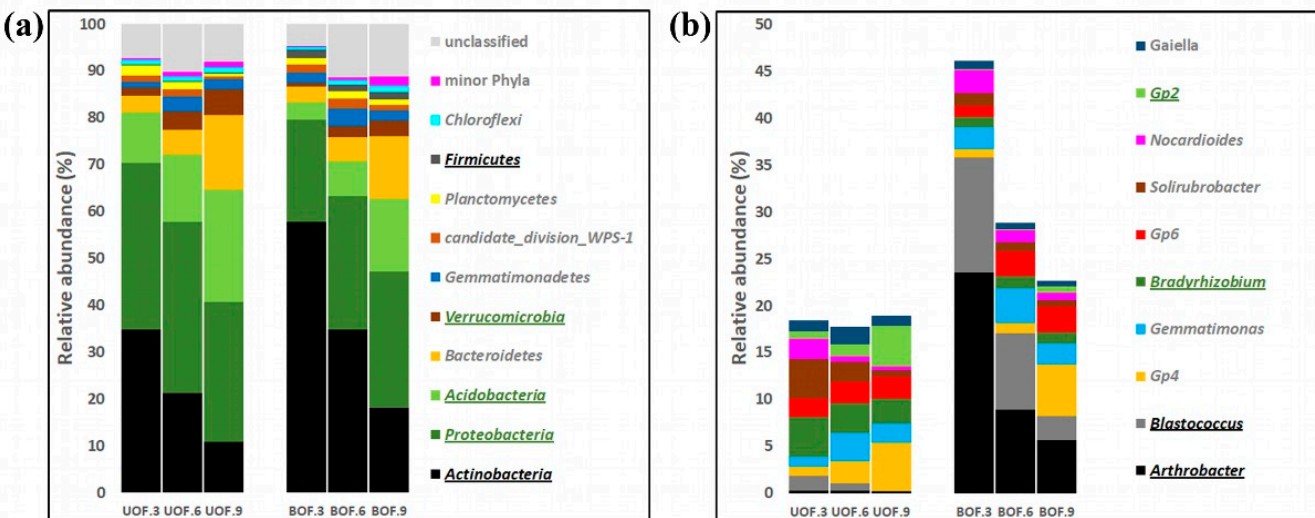

**Figure 2.** Taxonomical profiles at phylum (**a**) and genus (**b**) levels. Phyla lower than 1% in average relative abundance are merged as "minor Phyla". Genera lower than 1% in average relative abundance are not shown. Taxa with statistically significant higher abundance in unburned (UOF) and burned (BOF) holm oak forest rhizosphere soils are underlined and highlighted in bold green and bold black, respectively.

On the other hand, longitudinal comparisons showed that the taxonomic profile in the burned forest closely resembled the profile of the control at the final sampling time of 9 yaf. In fact, at the first two sampling times, *Actinobacteria* was the predominant phylum in the rhizosphere communities affected by fire (57.7% in BOF.3 and 34.7% in BOF.6), followed by *Proteobacteria* (21.9% and 28.6%, respectively). However, in the last sampling time, the main phylum in the BOF was *Proteobacteria*, the relative abundance of which was similar to that found in the UOF—29.1 and 29.9%, respectively.

Nonetheless, differences between burned and unburned forests were still evident when considering the main genera. This difference decreased over time, but was still noticeable at the final sampling time (Figure 2b). Furthermore, *Arthrobacter* and *Blastococcus*, which were the two main genera in the BOF, were significantly more abundant in the BOF at all times. In contrast, *Bradyrhizobium* and *Acidobacteria* group 2 (*Gp2*) maintained significantly higher relative abundances in the UOF over time (Figure 2b).

Further analysis of these results revealed 31 genera that had statistically significant differences between the burned and unburned forests (Supplementary Materials: Table S1). Interestingly, four of these (*Arthrobacter*, *Blastococcus*, *Massilia*, and *Microvirga*; Figure 3) showed a decrease in the BOF over time, although all but *Microvirga* remained significantly higher in the BOF than in controls (the UOF) 9 years after the fire. It is true that a decrease in the four above-mentioned genera was also observed in the unburned forest, but it was only noticeable and statistically significant in the case of *Blastococcus*. In contrast, *Gp2*, *Reyranella*, and *Rhizobacter*, which were statistically significantly more abundant 3 yaf in the UOF, showed an increase in the BOF according to time, which erased the statistical differences at 9 yaf in all the cited genera except *Rhizobacter*. On the other hand, *Mycobacterium* decreased in both conditions; it was significantly higher in relative abundance in the unburned forest than in the burned forest only at 3 yaf (Figure 3). Finally, the rest of the genera showed no clear trends over time (Supplementary Materials: Table S1).

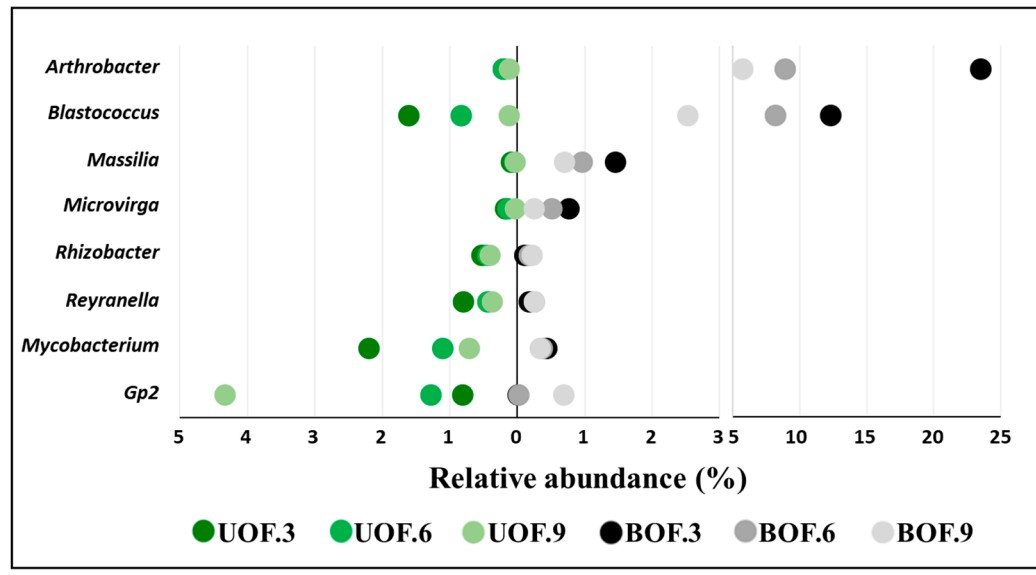

**Figure 3.** Mean relative abundances of the eight genera present in both conditions, with statistically significant differences and a decreasing or increasing trend over time in any condition. UOF = unburned holm oak forest, BOF = burned holm oak forest. The numerals 3, 6, and 9 denote sampling times in years after fire.

### 3.3. Soil Physicochemical Properties Drive the Microbial Taxonomic Profiles

The main differences between the burned and unburned forests in the parameters measured in the soil were pH and percentage of organic matter. Regarding longitudinal differences, the percentage of clay and available inorganic phosphorus (Pi) were the properties with the greatest alterations in both conditions (Table 2). Nevertheless, only pH and Pi showed a statistically significant effect ($p = 0.003$) on the prokaryotic profiles of the study communities (Figure 4). In fact, all microbial genera with differences between the UOF and the BOF, except *Microlunatus* and four minor genera, were statistically significantly and strongly correlated with pH (Supplementary Materials: Table S2).

Regarding pyrogenic carbon, 15 polycyclic aromatic hydrocarbons (PAHs) were measured at the initial and final sampling points (3 and 9 yaf). All of them were found at concentrations close to the detection level of the technique, suggesting that these compounds are rare in the studied forest rhizosphere soils, even after fire. Moreover, no statistically significant differences were found in any case between the burned and unburned rhizosphere. As for the longitudinal comparison, only naphthalene was statistically significantly reduced in the undisturbed forest over time ($T = 12.3$, $p = 0.003$) (Table 3).

**Table 2.** Physicochemical properties of the rhizosphere soils of unburned holm oak forest (UOF) and burned holm oak forest (BOF).

| Soil Variables | UOF | | BOF | |
|---|---|---|---|---|
| | 3 yaf | 9 yaf | 3 yaf | 9 yaf |
| Clay (%) | 21.00 ± 2.08 | 14.53 ± 1.46 | 20.50 ± 1.53 | 13.5 ± 0.78 |
| Sand (%) | 45.74 ± 5.51 | 46.91 ± 5.47 | 49.54 ± 6.51 | 47.36 ± 8.67 |
| Silt (%) | 33.26 ± 4.16 | 38.56 ± 4.09 | 29.96 ± 6.56 | 39.14 ± 9.20 |
| Textural class | Loam | | Loam | |
| pH (H$_2$O) | 6.10 ± 0.10 a | 5.77 ± 0.15 a | 7.60 ± 0.10 b | 7.23 ± 0.06 b |
| Available water (%) | 17.11 ± 2.52 | 15.22 ± 2.06 | 16.43 ± 3.75 | 19.52 ± 4.65 |
| EC [1] (mmhos/cm³) | 0.14 ± 0.04 | 0.12 ± 0.05 | 0.22 ± 0.03 | 0.19 ± 0.03 |
| Organic matter (%) | 7.61 ± 2.00 | 7.28 ± 2.90 | 4.54 ± 0.75 | 5.35 ± 0.69 |
| Total N (%) | 0.37 ± 0.17 | 0.37 ± 0.19 | 0.23 ± 0.04 | 0.30 ± 0.04 |
| C/N | 11.95 ± 3.69 | 11.58 ± 2.22 | 11.19 ± 0.28 | 10.17 ± 0.25 |
| Available Pi (ppm) | 8.00 ± 4.44 a | 23.00 ± 17.35 ab | 5.23 ± 0.83 a | 60.33 ± 14.05 b |
| K (ppm) | 445.23 ± 41.07 | 356.67 ± 41.63 | 330.12 ± 47.29 | 343.33 ± 47.26 |

[1] Electrical conductivity. Statistically significant differences across burn status and time are indicated by different letters. Only measures of the first time-point (3 years after fire, yaf) and the last time-point (9 yaf) are available. Means ± standard deviations.

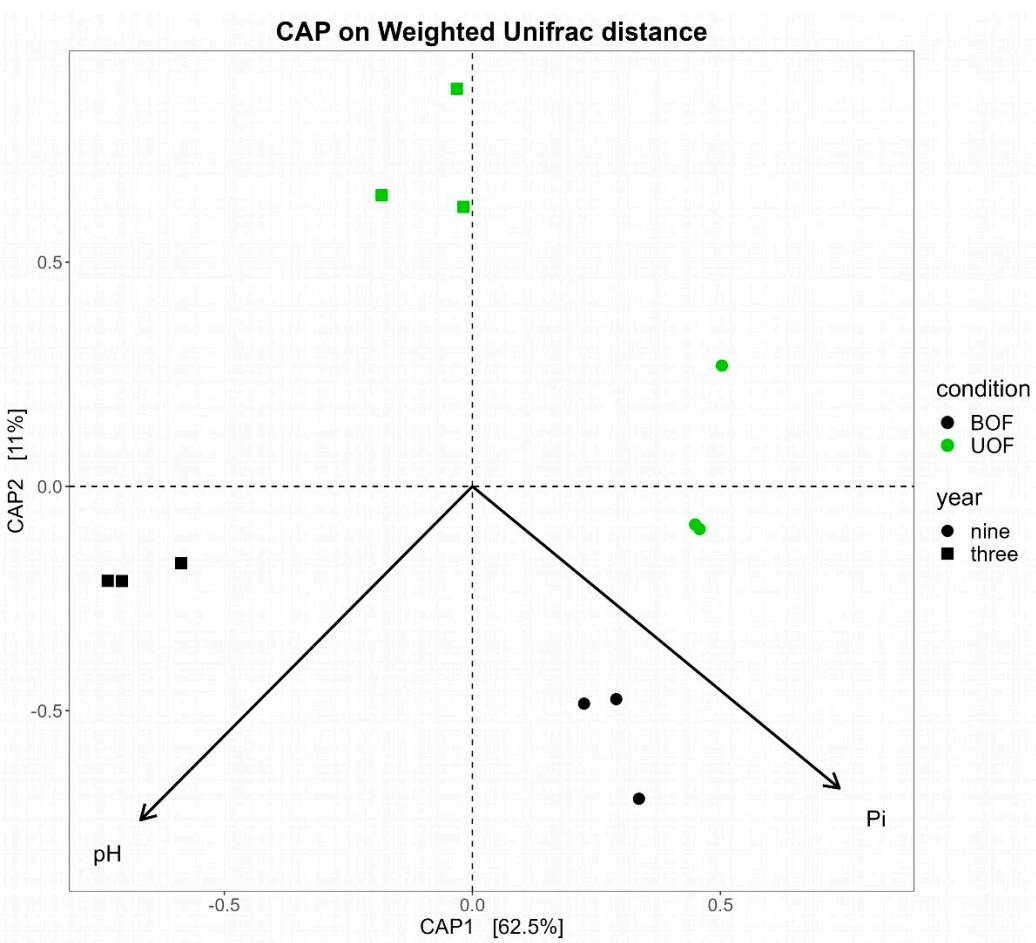

**Figure 4.** Constrained analysis of principal coordinates (CAP) of the rhizosphere soils of unburned holm oak forest (UOF) and burned holm oak forest (BOF). Only 3 and 9 yaf physicochemical data were available for this analysis. Vectors show only the parameters with a statistically significant effect on the prokaryotic communities.

### 3.4. Long-Term Alteration of Prokaryotic Biomass

Interestingly, higher prokaryotic biomass was observed in the burned holm oak forest rhizosphere soil at all three sampling times (*t*-test $T = -5.59$, $p < 0.001$). Although no statistically significant changes were found over time, a trend of increasing BOF biomass and decreasing UOF biomass was observed (Figure 5a). After quantifying the actinobacterial biomass it was also observed that the values in BOF were statistically significantly higher than in UOF ($T = -5.99$, $p < 0.001$; Figure 5a). However, the most remarkable alteration was observed when studying the ratio of actinobacterial *versus* total prokaryotic biomass. Indeed, the actinobacterial proportion was statistically significantly higher in BOF than in UOF ($T = -8.5$, $p < 0.001$) at the starting point (3 yaf), but declined to a similar proportion to those of control at subsequent sampling times (Figure 5b).

### 3.5. Satellite Monitoring of Fire

By satellite imaging, more than 50% of the pixels within the fire perimeter displayed moderate-to-very-high burn severity values (Figure 6). dNBR values of the three sampling points in the burned area ranged from $187 \pm 24$ to $413 \pm 27$ (mean $\pm$ std. error).

### 3.6. Post-Fire Vegetation Recovery

As expected, the NDVI of the burned plots sharply decreased after the fire (Figure 7). The NDVI of the 3 yaf images was significantly lower than pre-fire values ($p = 0.033$), but the 6 yaf and 9 yaf NDVI values did not differ significantly from pre-fire NDVI values

($p = 0.581$ and $p = 0.841$, respectively), which suggested a recuperation of the greenness 6 years after the fire.

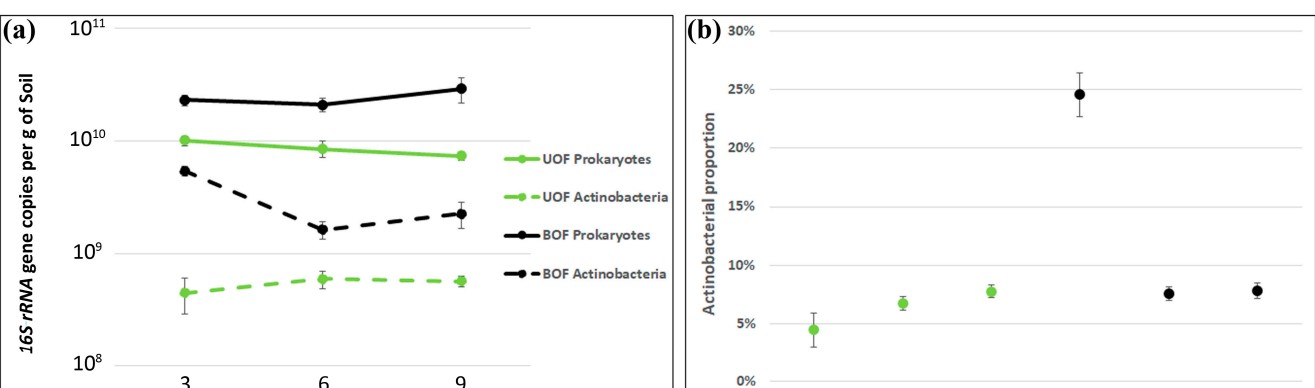

**Figure 5.** Prokaryotic and actinobacterial (**a**) biomass and their ratio (**b**) measured as *16S rRNA* gene copies per gram of soil. Data are shown in linear scale. Each point represents the average value of nine trees. Error bars represent the standard error.

*3.7. Strong Changes in Rainfall Regimes*

As this was a longitudinal study with a difference of 3 years between each sampling time, we analyzed the differences in the rainfall regime of each time interval. We studied the period from the beginning of the hydrological cycle up to the time of sampling. However, from our previous studies [47], we know that seasonal fluctuations greatly affect the microbial communities of the rhizosphere of woody plants, and that is why, in this analysis, we highlighted the rainfall accumulated in the 120 days prior to each sampling event (Figure 8). Regarding the hydrological cycle, the first sampling time was the driest and the second sampling time was the rainiest. However, when we focused on the 120 days prior to sampling, we still saw that the 2008 data had the least accumulated rainfall, but the 2014 data had the most accumulated rainfall (Figure 8).

**Table 3.** Soil polycyclic aromatic hydrocarbons (PAHs).

| | UOF | | BOF | |
|---|---|---|---|---|
| **PAHs** | **3 yaf** | **9 yaf** | **3 yaf** | **9 yaf** |
| Acenaphthene | $0.17 \pm 0.16$ | $0 \pm 0$ | $0.07 \pm 0.13$ | $0 \pm 0$ |
| Acenaphthylene | $0 \pm 0$ | $0.06 \pm 0.06$ | $0 \pm 0$ | $0.02 \pm 0.03$ |
| Anthracene | $2.65 \pm 0.65$ | $0.32 \pm 0.25$ | $2.45 \pm 1.35$ | $0.09 \pm 0.04$ |
| Benzo(g,h,i)perylene | $0.62 \pm 0.40$ | $0.33 \pm 0.28$ | $0.80 \pm 0.41$ | $0.16 \pm 0.04$ |
| Benzo-a-anthracene | $0.36 \pm 0.17$ | $0.19 \pm 0.08$ | $0.82 \pm 0.28$ | $0.16 \pm 0.03$ |
| Benzo-a-pyrene | $0.56 \pm 0.24$ | $0.25 \pm 0.16$ | $0.74 \pm 0.49$ | $0.12 \pm 0.03$ |
| Benzo-b,k-fluoranthene | $1.76 \pm 1.20$ | $0.86 \pm 0.46$ | $2.73 \pm 1.21$ | $0.62 \pm 0.14$ |
| Chrysene | $1.16 \pm 0.65$ | $0.47 \pm 0.26$ | $3.35 \pm 1.54$ | $0.63 \pm 0.36$ |
| Dibenzo(a,h)anthracene | $0.34 \pm 0.23$ | $0.3 \pm 0.22$ | $0.49 \pm 0.28$ | $0.10 \pm 0.03$ |
| Phenanthrene | $2.28 \pm 0.63$ | $0.82 \pm 0.07$ | $4.51 \pm 1.25$ | $0.57 \pm 0.13$ |
| Fluoranthene | $3.61 \pm 2.05$ | $1.76 \pm 1.02$ | $7.76 \pm 3.1$ | $0.88 \pm 0.16$ |
| Fluorene | $0.42 \pm 0.09$ | $0.16 \pm 0.02$ | $0.57 \pm 0.33$ | $0.12 \pm 0.02$ |
| Indeno(1,2,3-cd)pyrene | $0.34 \pm 0.23$ | $0.30 \pm 0.22$ | $0.51 \pm 0.27$ | $0.09 \pm 0.01$ |
| Naphthalene [1] | $1.15 \pm 0.07$ a | $0.56 \pm 0.04$ b | $2.18 \pm 0.98$ ab | $0.48 \pm 0.05$ b |
| Pyrene | $3.23 \pm 1.87$ | $1.54 \pm 0.86$ | $9.12 \pm 4.10$ | $0.90 \pm 0.31$ |

[1] Statistically significant differences across burn status and time are indicated by different letters. Measures of the first time-point (3 years after fire, yaf) and the last time-point (9 yaf) are shown. Means ± standard deviations of PAH concentrations are expressed in ppm.

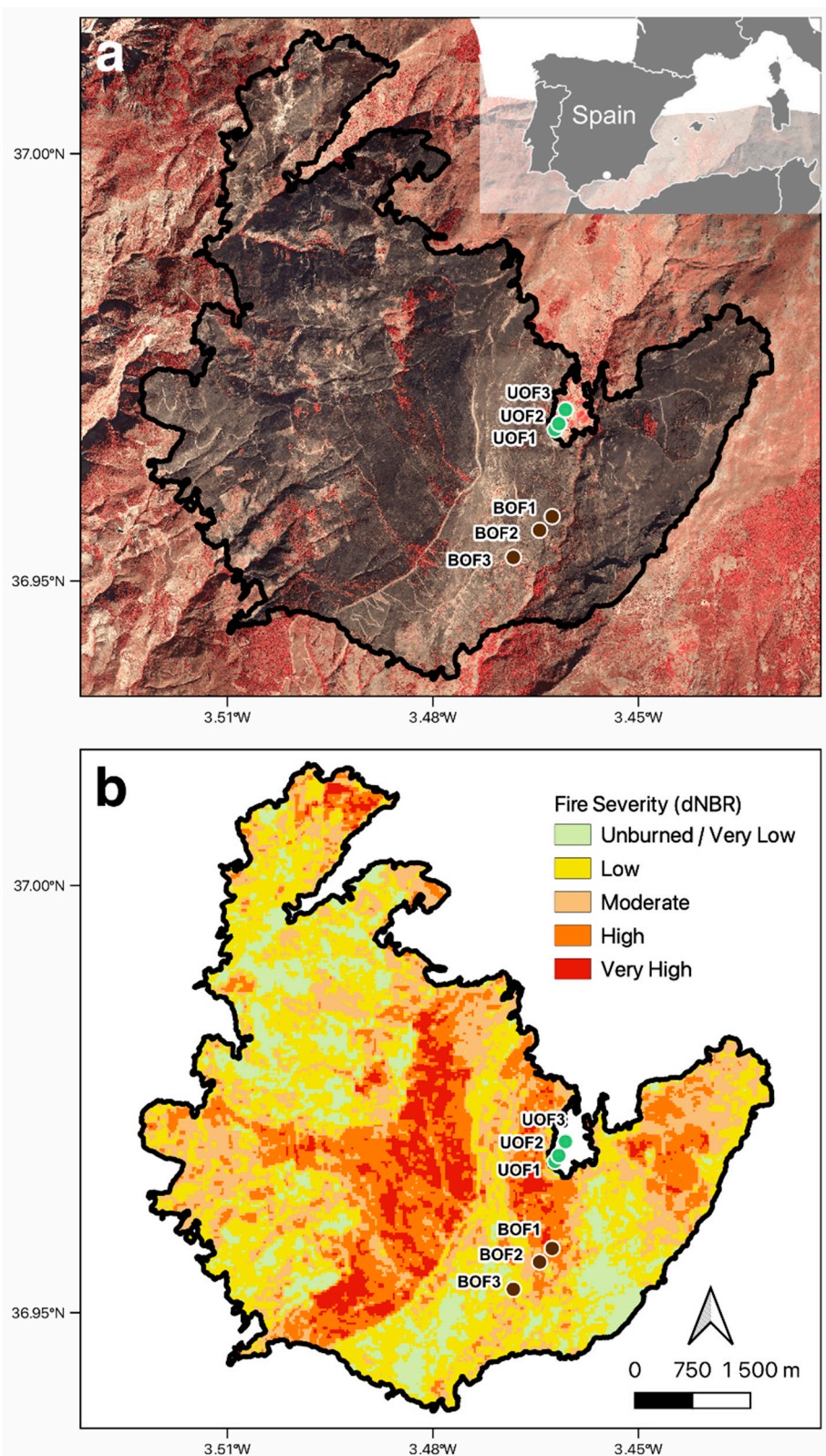

**Figure 6.** Location map of the Lanjarón fire. (**a**) Infrared ortophography with 0.25 m resolution generated from the digital photogrammetric flight of 06/10/2005 (source: Regional Government of Andalusia). (**b**) Fire severity map using the delta normalized burn ratio (dNBR), classified using the European Forest Fire Information Service (EFFIS) thresholds. Sampling points are shown.

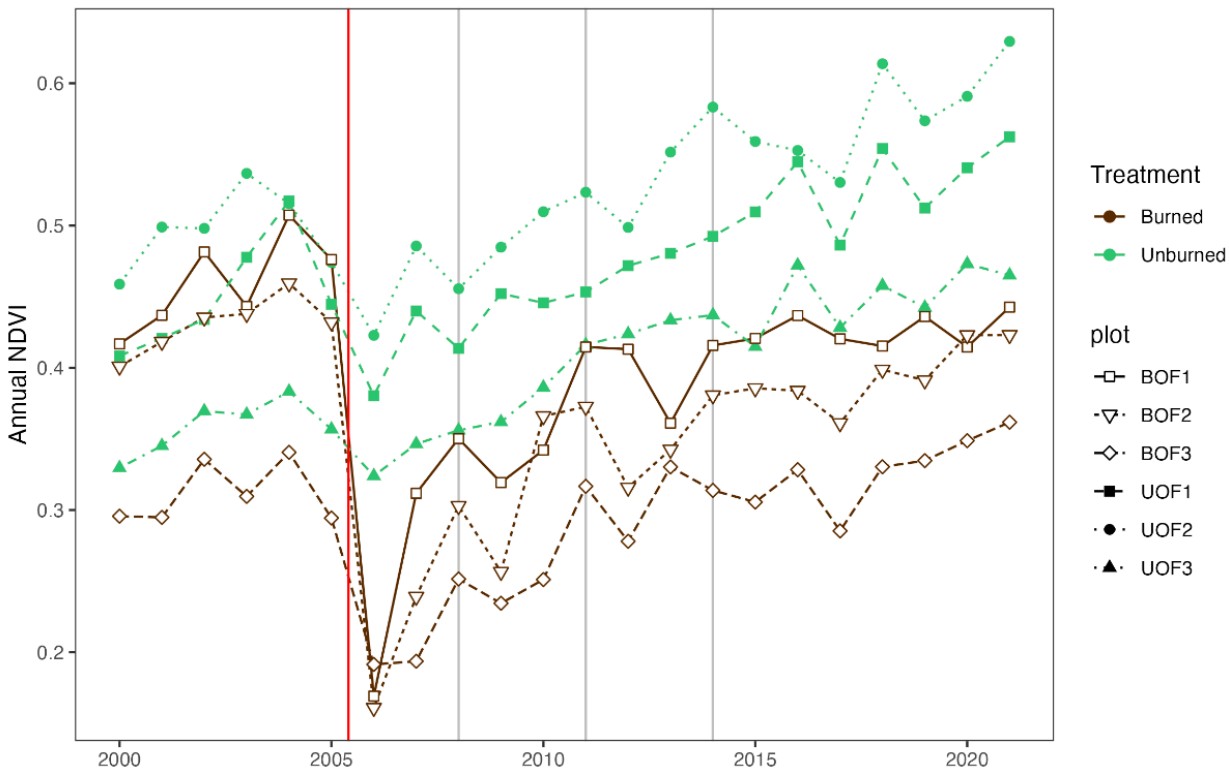

**Figure 7.** Temporal evolution of the annual NDVI for the sampling points. Vertical lines indicate the fire event (red line in September 2005) and the three sampling time points (2008, 2011, and 2014).

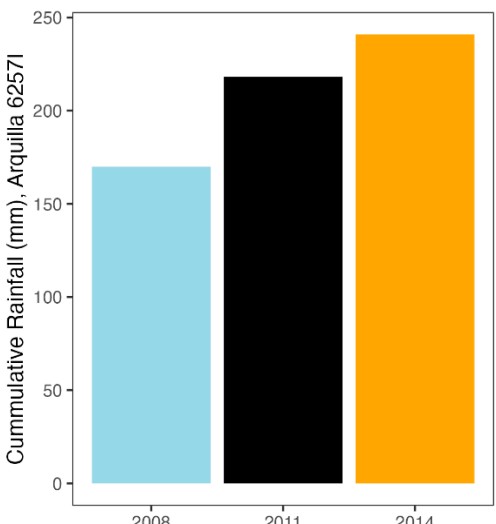

**Figure 8.** Cumulative rainfall in the 120 days prior to the sampling dates for each year.

## 4. Discussion

One of the most widely observed effects of a forest fire on microbial communities is a decrease in diversity. Our previous study showed a strong decrease in prokaryotic diversity 3 years after a fire [14]. Such a pattern is consistent with the findings of other cases that analyzed soil microbial communities at 1 yaf [11,12] and 3 yaf [15] in coniferous forests, as well as in Mediterranean shrublands at less than 3 yaf [22]. As shown in Figure 1a, in our study, this effect is caused mainly by a significant decrease in evenness or an increase in the dominance of a few species, and not so much by a loss of species (richness). This finding does not seem to be consistent with the findings of other studies. For instance,

in a study 3 months after a fire, a decrease in richness was observed in a mixed conifer forest, but not in a ponderosa pine forest [8]. Interestingly, in another study 3 months after a high-severity prescribed fire in a mixed aspen–fir forest [17], a decreasing trend in species richness was observed, which was higher in bulk soil than in rhizosphere, although it was not statistically significant in any case for the prokaryotic community. On the other hand, our results show an increase in prokaryotic diversity over time, which recovers the levels of the unburned area at 9 years after a fire. The same result has been observed in other studies, but at times later than a decade [20–22].

Regarding the recovery of the prokaryotic community structure and composition, the beta diversity analysis indicated that 9 years are not sufficient for observing the same structure between the burned and the control plots. These findings are in agreement with those obtained in a boreal conifer forest at 5 yaf [16], but do not allow us to confirm or refute that full recovery is achieved only after the first decade, as was observed through fingerprinting techniques in another boreal conifer forest in China at 11 yaf [21] and in a tropical forest in Indonesia at 9 yaf [19].

With respect to the taxonomic profiles and their evolution, a decrease in an abundant phylum, *Actinobacteria*, was observed over time in both conditions (the BOF and the UOF). This pattern may be related to a change in the rainfall regime, as the first sampling time was the driest, accounting for both the beginning of the hydrological cycle up to the sampling date of each year and the 4 months prior to each sampling event. In fact, a higher proportion of *Actinobacteria* is a sign of the more xeric conditions in a Mediterranean climate, as many species of this phylum are known to produce spores and/or to possess forms of resistance that allow them to withstand hostile conditions, such as drought or water limitation [9,48]. Nevertheless, the enrichment of this phylum in the burned area compared to the control was still evident in the last sampling time (9 yaf).

Some authors observed an increase in *Bacteroidetes* due to fire, together with increases in *Actinobacteria* and *Firmicutes* [8,11,15,21]. However, in our case, the phylum *Bacteroidetes* showed an increase over time, but there were no differences between the conditions. On the other hand, *Proteobacteria*, *Acidobacteria*, and *Verrucomicrobia* were negatively affected by the fire, a result that was consistent with the above-mentioned studies, although only the former recovered to the levels of the control soils at 9 yaf.

Finally, a connection between the rainfall regime and the temporal evolution of rhizosphere microbial communities was observed in both the burned and the unburned forests. Therefore, these differences may be associated with environmental changes, such as cumulative rainfall and other unmeasured changes. In fact, the temporal differences observed in beta diversity (for both the UOF and the BOF) may reflect both changes in the community (evolution toward a situation of stability, microbiome restoration) and environmental changes. The $R^2$ of the PERMANOVA showed that 25% of the variance remains unexplained by the factors we studied.

In our study, a finding that was in common with most published works on this subject was an enrichment of the genera *Arthrobacter* [8,10–12,15,16,49], *Blastococcus* [10–12,16,49] (*Actinobacteria*: *Micrococcaeae* and *Geodermatophilaceae*), and *Massilia* [10,11,16,49] (*β-Proteobacteria*, *Burkholderiales*) in burned soils. We suggest that a higher proportion of these three genera may be considered as a long-term biomarker of forest-fire-induced dysbiosis of the prokaryotic communities of the bulk and rhizosphere soils. Indeed, this disturbance extends for at least a decade and probably longer, since soil alkalization and water scarcity continue or, even worse in the current climate change scenario, increase over time.

On the one hand, wildfires exert a disturbance on soils, triggering significant changes in microbial communities. Pyrophilous microbes, as the three bacterial biomarkers highlighted here, possess traits such as heat-resistant and xerotolerant forms, affinity for nitrogen mineralization, and aromatic hydrocarbon degradation, which enable their survival in fire-impacted and post-fire environments [50]. Heat shock proteins and molecular chaperones further enhance thermal resistance. The presence of genes for mycothiol biosynthesis and

osmoprotectant synthesis (such as trehalose and glycine betaine) suggests adaptations for oxidative stress tolerance and maintaining cell viability in low soil moisture conditions post-fire. The combination of these traits appears to be an emergent property that supports the dominance of these taxa in fire-impacted environments.

Overall, these findings shed light on the mechanisms employed by pyrophilous microbes to thrive in fire-affected ecosystems in the short term [12]. Other authors indicated that after a forest fire, it is common to find an enrichment of copiotroph bacteria in the soil in the short term (1 yaf) [11,12] and the intermediate term (3 yaf) [15]. Members belonging to the three above-mentioned genera have been described as having this lifestyle [15,51,52]. Nevertheless, in a longer term, 5 yaf, this quality does not seem to be a determining factor in the community profile [16]. It is clear that pH is a key factor in the maintenance of dysbiosis, but Whitman et al. [16] suggested that pyrogenic organic matter (PyOM) could be another important factor. However, in our study, we did not detect recalcitrant forms of pyrogenic carbon in the burned soils at any of the sampling times. These findings are consistent with those recently published by Nelson et al. [12]. They detected an increase in PyOM in surface soils one year after fire, but not below 5–10 cm, which coincides with our sampling depth.

On the other hand, a factor that is also decisive in the maintenance of the fire-induced bacterial community alteration and the increase in their biomass is the fire severity [8,11,12]. For instance, despite both prokaryotic alpha diversity and vegetation NDVI values reaching pre-fire levels approximately a decade after the disaster, some microbial and physicochemical indicators did not, even in the plot where the fire severity was moderate. Nevertheless, all these indicators are slowly approaching the control values, reflecting the moderate-to-high severity of the fire. However, the prokaryotic biomass measured by the qPCR of the *16S rRNA* gene has not decreased over time. This finding is somewhat surprising, as it is strongly correlated with the increase in fast-growing bacteria one year after the fire [10,11], as they have a higher average copy number of this gene. Interestingly, the biomass of actinobacteria cannot explain this, as it returned to similar levels to that of the unburned condition in the medium term. Some of the mentioned authors stated that the greater the severity of the fire, the longer this indicator persists [11,12], although none of these studies analyzed a period as long as ours. In any case, the biomass in our burned forest did not seem to show a decreasing trend, almost a decade after the fire. Rather, it seemed to point in the opposite direction.

Finally, both the increase in pH and biomass and the alteration of the prokaryotic community (mainly the maintenance of high proportions of the three biomarkers, *Arthrobacter*, *Blastococcus*, and *Massilia*) are still evident almost a decade after fire in the rhizosphere soil of a Mediterranean holm oak forest. Therefore, we believe that longer-term monitoring is necessary for a full recovery of this ecosystem. In the future, we may find a similar community in both the control and the burned conditions or, on the contrary, it is possible that the disturbance has completed the establishment of a new post-fire microbial community.

## 5. Conclusions

Previous studies have shown a consistent pattern of decreased prokaryotic diversity in the aftermath of fires, primarily due to a decrease in evenness rather than species richness. However, there are some inconsistencies in the findings of different studies, indicating that the effects of fire on microbial diversity can vary, depending on factors such as fire severity and ecosystem type. It is worth noting that the recovery of a prokaryotic community structure and composition takes time; even after 9 years, there are still discernible differences between burned and unburned areas. For instance, the enrichment of specific genera, such as *Arthrobacter*, *Blastococcus*, and *Massilia*, in burned soils may serve as long-term biomarkers of fire-induced dysbiosis.

In summary, wildfires lead to significant alterations in soil microbial communities, with certain bacterial taxa thriving in fire-impacted environments due to their unique adaptive traits. Pyrophilous microbes, characterized by heat resistance, affinity for nitrogen

mineralization, and aromatic hydrocarbon degradation, demonstrate their ability to survive and dominate in post-fire environments. The presence of specific genes and adaptations for harsh conditions further support their resilience. It is possible that a new microbial community has been established after a fire, which may differ from the pre-fire state. Finally, the persistence of altered community composition, biomass, and pH suggests that longer-term monitoring is necessary to assess the full recovery of fire-affected ecosystems.

**Supplementary Materials:** The following supporting information can be downloaded at: https://www.mdpi.com/article/10.3390/f14071383/s1, Figure S1: Map of the sampling area; Figure S2: PCR to check actinobacterial primers specificity; Table S1: Mean relative abundance at genus level.; Table S2: Spearman correlations of pH and Pi versus the prokaryotic community at genus level.

**Author Contributions:** Conceptualization, M.F.-L.; methodology, M.F.-L., P.J.V., J.F.C.-D. and A.J.F.-G.; data curation, A.J.F.-G.; formal analysis, A.J.F.-G., A.V.L., F.M.G.-R. and A.J.P.-L.; investigation, M.F.-L. and A.J.F.-G.; resources, P.J.V. and J.F.C.-D.; writing—original draft preparation, A.J.F.-G.; writing—review and editing, A.J.F.-G., A.V.L. and M.F.-L.; visualization, A.J.F.-G.; supervision, M.F.-L.; project administration, M.F.-L.; funding acquisition, M.F.-L. and S.G.T. All authors have read and agreed to the published version of the manuscript.

**Funding:** This research was funded by the following grants: P08-CVI-03549 from The Department of Innovation, Science and Enterprise of the Autonomous Government of Andalusia; OAPN 021/2007 from The National Parks Autonomous Body (Ministry of the Environment) and 20134R069, RECUPERA 2020 from the Spanish Ministry of Economy and Competitiveness and CSIC, including the European Regional Development Fund (ERDF). The work (10.46936/10.25585/60007435) conducted by the U.S. Department of Energy Joint Genome Institute (https://ror.org/04xm1d337 (accessed on 1 May 2023)), a DOE Office of Science User Facility, was supported by the Office of Science of the U.S. Department of Energy under Contract No. DE-AC02-05CH11231. This study was also funded by the Ministry of Science and Innovation of Spain through the European Regional Development Fund [SUMHAL, LIFEWATCH-2019-09-CSIC-4, POPE 2014-2020].

**Data Availability Statement:** The datasets generated and analyzed during the current study are available in the NCBI Sequence Read Archive (SRA) under BioProject number PRJNA291009.

**Conflicts of Interest:** The authors declare no conflict of interest.

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
