# Peer review of "Long-Term Persistence of Three Microbial Wildfire Biomarkers in Forest Soils"

_forests, doi:10.3390/f14071383_

Round 1
Reviewer 1 Report
GENERAL COMMENTS
Sound study with a wealth of analysis, some of which isn’t needed in the context of the posed research Qs. The discussion in turn poses mechanisms for observed trends without having conducted formal analyses (e.g., role of rainfall). I would encourage the authors to include fire severity and rainfall in their formal analyses of changes in microbial communities and consider also revisiting the introduction and research questions in the context of additional material that was explored (fire severity, vegetation recovery, rainfall). If data was available for 1 year post fire then this should be included here. Detailed comments follow.
TITLE
I suggest that 9 years is not ‘long-term’, and that descriptions such as ‘short’ and ‘long’ are rarely useful, but if used should be in the context of the life span of the ecosystem under study. Given the much longer life span of holm oak forests, the study would be better served by direct reference to the timeframe of the study, and the ecosystem under consideration.
ABSTRACT
Start with the key knowledge gap – the two starting sentences don’t add much value and some would argue that 9 years post-fire is rather short-term. Key knowledge gap around temporal recovery of microbial communities following wildfire would be a stronger starting point.
L18 ‘To perform this study’, not needed – aim to be more direct (less passive language)
L19 ‘In this research it has been observed’ – again passive (I won’t continue highlighting passive language – I’m sure you get what I mean)
Needs to be clearer that the three genera increased in response to wildfire and have not returned to ‘control’ conditions (assuming this is long-term unburnt forest – but wasn’t defined). What is the functional significance of these genera?
Readers will also be asking why focus on these genera and did other taxa increase?
INTRODUCTION
Opening paragraph – a description of the extent (i.e., hectares) of this forest (in Spain) would help with context
L46-48, very difficult to follow, do the numbers represent fire events per decade and if so it’s hard to relate the numbers to particular time periods – please simplify.
L56, please qualify what you mean by ‘short’ and ‘intermediate’ (i.e., provide time frames). Better to start paragraph with messaging on importance of microbial communities in forest ecosystems before moving into length of studies (i.e., why it’s important to study them in the first place).
L61, not clear why you mean by Arthrobacter role in fire restoration – is nitrogen limitation particularly problematic for oak forest? More needed here.
L72, revisit terminology in relation to length of study – indication of years is sufficient.
In relation to the study objectives, more needed in introduction regarding the importance of rhizosphere communities in ecosystem recovery to disturbance and the role of carbon in microbial community composition and dynamics
MATERIALS AND METHODS
L85, round off to nearest hectare
L88, ‘deeply’ affected meaning?
International readers would appreciate a map of the study area and relative locations of burnt / unburnt sampling locations [I see now this is presented in Figure 6 – which should be shown up front]
If the authors were involved with the first year analysis, then I would suggest this data should be included in this study to provide an extra point (1 year) in relation to temporal trends.
L96, direction of transect – along or adjacent to slope contour?
L97, useful to spell out sampling intensity (2 sites x 3 plots/site x3 trees/plot = 18 trees)
L103, were these trees the same as those sampled 1 year after fire?
L106, ‘per condition’, meaning? [I see this is described later, L144, but should be clear up front]
L108, ‘etc.’ is meaningless, describe the soil variables measured
L109, reference for these standardized procedures please
L156, A SIMPER test following the PERMANOVA would allow you to assess which taxa made the greatest contributions to observed differences among condition / time points.
L167, no need to state positive or negative, refer to absolute value of 0.6 as your criteria for significant correlation.
I found the move from statistical analysis back into variable measurement confusing (section 2.5, 2.6, 2.7). I would also expect that fire severity was conducted prior to site selection and that sites were established in ‘very high’ severity class? This needs clarity and tie in with site selection criteria.
Not clear how section 2.8 and 2.9 related to the research Qs. The NDVI would be representative of the entire vegetation cover – while collected soil was limited to oak rhizosphere associated soil (rather then composite ‘general’ soil representative of the broader vegetation community). If climate differed among sampling periods (as you might expect) – then I would expect the climate data to be incorporated into the statistical models – but it looks like this isn’t the case?
RESULTS
L248, ‘almost recovered’ is not very meaningful – something like ‘temporal recovery in microbial diversity to 9 years after fire’
Figure 1, resolution is poor, use of letters to show pair-wise differences in figures would create less clutter (bust this is personal preference I guess)
L251, forest status or forest condition or situation? Consistency needed.
L266-275, clarity needed around the PERMANOVA results – no interaction effects for burn status x time? Pairwise differences were limited to 9 vs (3 and 6), no differences 3 v 6? Perhaps a table of PERMANOVA results as a supplement? This is also where a SIMPER would be useful to show which taxa contributed to observed differences [I guess this depends on how informative the OTU data is].
L276-315, a PERMANOVA examining burn status, time and their interaction could also be applied to the phylum and genus level data and from there SIMPER used to assess which taxa contributed to observed significant differences; this would be more statistically defensible than individual tests per taxa (in my view), then depending on the PERMANOVA outcomes, figures could represent burnt status, time, or burn status and time (shown currently – but really only needed if there is significant interaction)
Figure 2, so underlined green is > in unburned, and black is > in burnt (needs to be clearer).
Figure 3, very difficult to interpret; couldn’t data simply be burnt relative to unburnt to show increase or decrease over time (i.e. relative to control)?
L324, but Table 2 doesn’t indicate significant differences in organic matter, same for clay and Pi? If there are statistical differences then Table 2 needs to be revisited in terms of how pairwise differences are shown.
Table 2, ‘Samples’ title not needed; no samples taken at 6 years? pH means and errors need to be recalculated – need to be mindful that pH = -log10[H+], so to determine mean pH, need to first determine mean [H+] then transform to pH, errors for pH aren’t meaningful and best approach is to report 95% confidence intervals for pH. This is a very common error in reporting of pH means and errors. And letters represent differences across burn status and time – need clarity here. Additional columns could be included to the table to show significance tests for burn status, time and interaction effects.
Table 3, letters are across burn and time? No need for ‘Samples’ title
L340, no interaction effects?
L367-374, I would expect this type of information to be used for site selection, rather than presented in the results sections (given also no research Q related to this information).
Figure 7, should indicate in figure the time of the fire. Further, not clear how this information feeds into the study Qs? Remove.
L384-394, all very interesting, but data could be better used in the analysis, rather than purely descriptive.
DISCUSSION
L421, this is not supported by the results (the CAP), which included available water, but for some reason did not include the data of the rainfall. I suggest the rainfall data is included in the CAP analysis
L430, new paragraph
L436-439, reason again to include rainfall data as a predictor variable in the analyses (e.g. in the CAP).
L449, were bulk soils assessed? I thought from the methods only rhizosphere soils were collected?
L463, this discussion is out of place given lack of research Q; further fire severity data would have served useful in site selection. Opportunity to include fire severity and vegetation recovery in formal analysis and include a formal research Q around fire severity, vegetation recovery.
L479-486, closing paragraph a little weak, future monitoring is always important.
Some passive language - further details provided in other comments - nothing of significant concern
Author Response
Reviewer 1
Comments and Suggestions for Authors
GENERAL COMMENTS
Sound study with a wealth of analysis, some of which isn’t needed in the context of the posed
research Qs. The discussion in turn poses mechanisms for observed trends without having
conducted formal analyses (e.g., role of rainfall). I would encourage the authors to include fire
severity and rainfall in their formal analyses of changes in microbial communities and consider
also revisiting the introduction and research questions in the context of additional material that
was explored (fire severity, vegetation recovery, rainfall). If data was available for 1 year post fire
then this should be included here. Detailed comments follow.
TITLE
I suggest that 9 years is not ‘long-term’, and that descriptions such as ‘short’ and ‘long’ are rarely
useful, but if used should be in the context of the life span of the ecosystem under study. Given
the much longer life span of holm oak forests, the study would be better served by direct reference
to the timeframe of the study, and the ecosystem under consideration.
Reply: In the title we want to point out that these microbial biomarkers appear in other
forest soils and not only in the holm oak forest. Moreover, they persist in the long term.
According to the cited literature in this study, 1 year or less is short term (Nelson et al.
2022 Nature Microb. Ref: 12; less than 3 years after fire according to Whitman et al. 2022
SBB Ref: 16), 3 to 5 years is intermediate term (Adkins et al. 2020 STOTEN Ref: 15. This
description is used in the title) and longer is considered long-term period in some postfire
microbial community studies (Isobe et al. 2009 JGAM Ref: 19; Xiang et al. 2014 Ref:
21). Furthermore, the monitoring carried out can be considered long-term from the point
of view of the microbial communities’ evolution.
ABSTRACT
Start with the key knowledge gap – the two starting sentences don’t add much value and some
would argue that 9 years post-fire is rather short-term. Key knowledge gap around temporal
recovery of microbial communities following wildfire would be a stronger starting point.
Reply: According to the reviewer comment we have removed the first two sentences of
Abstract and added “Long-term monitoring of microbial communities in the rhizosphere
of post-fire forests is currently one of the key knowledge gaps.” Moreover, the expression
long-term is explained in our first answer.
L18 ‘To perform this study’, not needed – aim to be more direct (less passive language)
Reply: It was changed according to the reviewer comment.
L19 ‘In this research it has been observed’ – again passive (I won’t continue highlighting passive
language – I’m sure you get what I mean)
Reply: It was changed according to the reviewer comment. But passive or active is a point
of literature style that does not affect the results or the science of the manuscript.
Needs to be clearer that the three genera increased in response to wildfire and have not returned
to ‘control’ conditions (assuming this is long-term unburnt forest – but wasn’t defined). What is the
functional significance of these genera?
Reply: We think this sentence in the abstract “The relative abundance of these biomarkers
shows a decreasing trend over time, but are still far from the values of the control
condition” gives a clear message about what is required in this reviewer comment.
To highlight the functional significance of these genera we add to the abstract the next
sentence “These pyrophilous microbes possess remarkable resilience against adverse
conditions, exhibiting traits such as xerotolerance, nitrogen mineralization, degradation of
aromatic compounds, and copiotrophy in favorable conditions. Furthermore, these
biomarkers thrive in alkaline environments, which persist over the long term following
forest fires”. We add a new paragraph in discussion to expand these ideas.
Readers will also be asking why focus on these genera and did other taxa increase?
Reply: This issue was analyzed in more detail in our previous study, based only on data 3
years after the fire (Fernández-González et al. 2017 Scientific Reports Ref: 14)
In addition, line 346 (after Figure 2) states that we found 31 genera with statistically
significant differences and are detailed in Table S1. We focused on those that had a clear
downward trend over time after enrichment by fire at the first sampling time (Arthrobacter,
Blastococcus, Massilia and Microvirga). The last one was eliminated as a long-lasting
biomarker because it recovered its relative abundance to control levels 9 years after the
fire. Moreover, in the discussion section it is mentioned that other authors described the
same bacterial genera as important ones after a wildfire.
We added this part of the sentence “and with no record of fires in the last century.” In
section 2.1 to better specify the conditions of the control situation.
INTRODUCTION
Opening paragraph – a description of the extent (i.e., hectares) of this forest (in Spain) would help
with context
Reply: We included a sentence according to the reviewer comment in opening paragraph.
L46-48, very difficult to follow, do the numbers represent fire events per decade and if so it’s hard
to relate the numbers to particular time periods – please simplify.
Reply: We have modified this paragraph according to the reviewer comment.
L56, please qualify what you mean by ‘short’ and ‘intermediate’ (i.e., provide time frames). Better
to start paragraph with messaging on importance of microbial communities in forest ecosystems
before moving into length of studies (i.e., why it’s important to study them in the first place).
Reply: These terms have been described and a paragraph added according to reviewer
comment.
L61, not clear why you mean by Arthrobacter role in fire restoration – is nitrogen limitation
particularly problematic for oak forest? More needed here.
Reply: We do not think it is necessary to elaborate further on this point as it is not within
the main issues of this work and was discussed extensively in our previous study (Cobo-
Díaz et al. 2015 Microb Ecol Ref: 6).
L72, revisit terminology in relation to length of study – indication of years is sufficient.
Reply: We believe that we have already given substantial reasons for maintaining this
terminology that is the usual for specialist author in this subject.
In relation to the study objectives, more needed in introduction regarding the importance of
rhizosphere communities in ecosystem recovery to disturbance and the role of carbon in microbial
community composition and dynamics
Reply: We added a paragraph in line 56 (now line 58) according to the reviewer
recommendations.
MATERIALS AND METHODS
L85, round off to nearest hectare
Reply: Modified according to the reviewer comment.
L88, ‘deeply’ affected meaning?
Reply: As can be seen in Figure 6b, the samples collected in the burnt holm oak forest are
in one of the areas where the fire was most severe. Hence, we say deeply affected to
emphasize that the fire affected them completely and not marginally or transiently.
Anyway, we changed to “severely Impacted by the wildfire”.
International readers would appreciate a map of the study area and relative locations of burnt /
unburnt sampling locations [I see now this is presented in Figure 6 – which should be shown up
front]
Reply: A map of the sampling area is added as Supplementary Material: Figure S1 and
indicated in section 2.1 according to the reviewer comment.
If the authors were involved with the first year analysis, then I would suggest this data should be
included in this study to provide an extra point (1 year) in relation to temporal trends.
Reply: The first sampling time was almost 3 years after the fire. Unfortunately, there is no
data prior to this date since there was not financial support in the first year after the fire
neither authorization to sample these sites.
L96, direction of transect – along or adjacent to slope contour?
Reply: We chose a transect to maintain the altitude, more or less, of the slope.
L97, useful to spell out sampling intensity (2 sites x 3 plots/site x3 trees/plot = 18 trees)
Reply: Information added according to the reviewer comment.
L103, were these trees the same as those sampled 1 year after fire?
Reply: The first sampling time was 3 years after fire. Yes, we sampled the same 18 trees 3,
6 and 9 year after fire to perform a true longitudinal study, as explained all trees were
marked by GPS and green paint.
L106, ‘per condition’, meaning? [I see this is described later, L144, but should be clear up front]
Reply: Added (burn status) to clarify this meaning according to the reviewer comment.
L108, ‘etc.’ is meaningless, describe the soil variables measured
Reply: Changed according to the reviewer comment.
L109, reference for these standardized procedures please
Reply: This methodology is available in supplementary material Figure S1 from Cobo-Díaz
et al. 2015 Microb Ecol (Ref: 6) and in supplementary information from Fernández-González
et al. 2017 Scientific Reports (Ref: 14).
L156, A SIMPER test following the PERMANOVA would allow you to assess which taxa made
the greatest contributions to observed differences among condition / time points.
Reply: We appreciate the suggestion, but we have avoided using SIMPER because the
developers of the package themselves say that " The results of simper can be very difficult
to interpret. The method very badly confounds the mean between group differences and
within group variation, and seems to single out variable species instead of distinctive
species (Warton et al. 2012). Even if you make groups that are copies of each other, the
method will single out species with high contribution, but these are not contributions to
non-existing between-group differences but to within-group variation in species
abundance." (https://rdrr.io/rforge/vegan/man/simper.html)
L167, no need to state positive or negative, refer to absolute value of 0.6 as your criteria for
significant correlation.
Reply: Modified according to the reviewer comment.
I found the move from statistical analysis back into variable measurement confusing (section 2.5,
2.6, 2.7). I would also expect that fire severity was conducted prior to site selection and that sites
were established in ‘very high’ severity class? This needs clarity and tie in with site selection
criteria.
Reply: Reviewer is right. We modified the title of section 2.4 to “Statistical analyses from
NGS data” as each subsequent section has its own detailed statistics. We noticed that
information was missing in 2.5 and we added a sentence with the statistical procedure.
The burnt holm oak forest was a mixed forest with holm oaks, oaks and conifers. In
addition, an experimental reforestation with redwoods and Pinus arizonica was carried out
in the 1930s. That is why, after the fire, holm oaks were selected in accessible areas and if
they had resprouted (those that died completely were not considered in these studies).
Thus, these were the selection criteria for drawing the sampling transect.
I would like to remind the reviewer that this methodology has already been published in
two previous papers (Ref: 6 and 14).
Not clear how section 2.8 and 2.9 related to the research Qs. The NDVI would be representative
of the entire vegetation cover – while collected soil was limited to oak rhizosphere associated soil
(rather than composite ‘general’ soil representative of the broader vegetation community). If
climate differed among sampling periods (as you might expect) – then I would expect the climate
data to be incorporated into the statistical models – but it looks like this isn’t the case?
Reply: Our study focuses on the prokaryotic communities of the holm oak rhizosphere (the
dominant woody community) and not on the entire plant community of the forest under
study. It is true that NDVI cannot discern at the scale of a single individual, but it gives us
an indication of the degree of recovery of each area sampled after the fire. By including
this index, we want to show that, although the forest stand has recovered at the
macroscopic level in the last years sampled (6 and 9 yaf), the microbial communities and
the physicochemical parameters of the soil influenced by the dominant plant species
(Quercus ilex) are still altered.
We could not correlate the accumulated rainfall data as we only have data from one
meteorological station, which prevents us from having averages and statistics with these
values. That is why in the discussion we talk about connection and not correlation and say
that this factor could have an effect on the communities without asserting such an
interaction. This is what we say in the end of the 3rd paragraph of discussion “Finally, a
connection between the rainfall regime and the temporal evolution of rhizosphere
microbial communities is observed in both burned and unburned forests. Therefore, these
differences may be associated with environmental changes such as cumulative rainfall
and other unmeasured changes. In fact, the temporal differences observed in beta
diversity (for both UOF and BOF) may reflect both changes in the community (evolution
towards a situation of stability, microbiome restoration) and environmental changes. The
R2 of the PERMANOVA shows that 25% of the variance remains unexplained by the factors
we have studied.”
RESULTS
L248, ‘almost recovered’ is not very meaningful – something like ‘temporal recovery in microbial
diversity to 9 years after fire’
Reply: Modified according to the reviewer suggestion.
Figure 1, resolution is poor, use of letters to show pair-wise differences in figures would create
less clutter (bust this is personal preference I guess)
Reply: Thanks for the suggestion, but we prefer to leave the image as it is, its quality is
600 dpi, and we think that the degree of significance of the p-value is informative in this
case.
L251, forest status or forest condition or situation? Consistency needed.
Reply: Situations was replaced by conditions in line 293. We think that always using
condition or forest status is very redundant and we have checked that it is clear what we
mean whenever we talk about conditions ("burned" and "control").
L266-275, clarity needed around the PERMANOVA results – no interaction effects for burn status
x time? Pairwise differences were limited to 9 vs (3 and 6), no differences 3 v 6? Perhaps a table
of PERMANOVA results as a supplement? This is also where a SIMPER would be useful to show
which taxa contributed to observed differences [I guess this depends on how informative the OTU
data is].
Reply: Yes, there was an interaction between forest status and time, P = 0.01. This
sentence was added at the end of the paragraph “Finally, was detected significant
interaction between both factors (PERMANOVA test P = 0.01; R2 = 0.09), which justifies
their separation in subsequent section plots.”. Reviewer is right, there was no statistically
significant difference between 3 and 6 yaf. Remember, as it says in the first sentence of
this paragraph, the 3 times were compared by pooling the data from both conditions
(burned and control). Ideally, it would be better to compare the 3 times for each condition
separately. In fact, this is indicated by the fact that there is interaction between the 2
factors. Unfortunately, with 3 replicates for each time and forest status it is not possible
to obtain statistically significant results with pairwise Adonis. We understand that this is
a constraint in our study, but we think that the PCoA together with PERMANOVA shows
that there is a clear approximation between the burned and control forest rhizosphere
microbial communities over time. We have argued against the use of SIMPER in a previous
comment.
L276-315, a PERMANOVA examining burn status, time and their interaction could also be applied
to the phylum and genus level data and from there SIMPER used to assess which taxa contributed
to observed significant differences; this would be more statistically defensible than individual tests
per taxa (in my view), then depending on the PERMANOVA outcomes, figures could represent
burnt status, time, or burn status and time (shown currently – but really only needed if there is
significant interaction)
Reply: We have not seen that PERMANOVA and PCoA at different taxonomic levels added
anything new to the results already presented. We have argued against the use of SIMPER
in a previous comment.
Figure 2, so underlined green is > in unburned, and black is > in burnt (needs to be clearer).
Reply: We see clear this sentence “Taxa with statistically significant higher abundance in
unburned (UOF) and burned (BOF) holm oak forest rhizosphere soils are underlined and
highlighted in bold green and bold black, respectively”
Figure 3, very difficult to interpret; couldn’t data simply be burnt relative to unburnt to show
increase or decrease over time (i.e. relative to control)?
Reply: Although the reviewer's idea seems a good choice, we know that similar plots, albeit
with bars instead of dots, are common in such publications comparing relative
abundances of microorganisms under two different conditions. We have therefore decided
to keep the figure.
L324, but Table 2 doesn’t indicate significant differences in organic matter, same for clay and Pi?
If there are statistical differences then Table 2 needs to be revisited in terms of how pairwise
differences are shown.
Reply: We performed these pairwise comparison with t_test function from rstatix R
package. I have re-analyzed the data and the only parameters with statistically significant
differences are pH and Pi.
Table 2, ‘Samples’ title not needed; no samples taken at 6 years? pH means and errors need to
be recalculated – need to be mindful that pH = -log10[H+], so to determine mean pH, need to first
determine mean [H+] then transform to pH, errors for pH aren’t meaningful and best approach is
to report 95% confidence intervals for pH. This is a very common error in reporting of pH means
and errors. And letters represent differences across burn status and time – need clarity here.
Additional columns could be included to the table to show significance tests for burn status, time
and interaction effects.
Reply: Samples removed. Physicochemical parameters were measured by a specialized
external service. For pH, the company provided us with a value, based on the
quantification of protons in H2O, for each soil sample submitted and we calculated the
mean and standard deviation of these values (n = 3) to show them in Table 2. Here we have
two groups per factor and we performed pairwise t tests for analyzing these data, due to
that there is no interaction effect like that of ANOVA tests.
Table 3, letters are across burn and time? No need for ‘Samples’ title
Reply: Modified according to the reviewer comment.
L340, no interaction effects?
Reply: We did the pairwise comparison directly with the t_test function, so the interaction
of the two factors was not studied. However, as indicated in the text, the values were very
close to the minimum detection level and there was nothing significant except for
naphthalene. And this showed no difference between burned and control, which is what
was initially aimed for.
L367-374, I would expect this type of information to be used for site selection, rather than
presented in the results sections (given also no research Q related to this information).
Reply: We have added a figure in supplementary material with this information as
requested in a previous comment. These images allowed us to obtain the dNBR. Although
not stated as one of the main objectives we think it supports the results discussed in the
penultimate paragraph of discussion section.
Figure 7, should indicate in figure the time of the fire. Further, not clear how this information feeds
into the study Qs? Remove.
Reply: Plot corrected according to first reviewer comment. We have justified the use of
NDVI in a previous comment.
L384-394, all very interesting, but data could be better used in the analysis, rather than purely
descriptive.
Reply: As we said in a previous comment, these data are from a single meteorological
station and we have no statistics to make robust correlations with them, but they do serve
as an indication of a trend.
DISCUSSION
L421, this is not supported by the results (the CAP), which included available water, but for some
reason did not include the data of the rainfall. I suggest the rainfall data is included in the CAP
analysis
Reply: The reviewer is right about available water. We modified the sentence to “This
pattern can be related to a change in rainfall regime,”. It is not correct to incorporate
cumulative rainfall data into the CAP analysis as all replicates for each year and forest
status would have the same repeated value as this parameter has no replicates.
L430, new paragraph
Reply: Modified according to the reviewer comment.
L436-439, reason again to include rainfall data as a predictor variable in the analyses (e.g. in the
CAP).
Reply: Already answered in a previous comment.
L449, were bulk soils assessed? I thought from the methods only rhizosphere soils were
collected?
Reply: In our study only rhizosphere samples were analyzed, but these biomarkers are
found in bulk soils presented in other studies cited in the discussion.
L463, this discussion is out of place given lack of research Q; further fire severity data would have
served useful in site selection. Opportunity to include fire severity and vegetation recovery in
formal analysis and include a formal research Q around fire severity, vegetation recovery.
Reply: We have already justified in a previous comment the sampling methodology. We
believe that fire severity is not a key factor in our design, since samples were not taken at
the same level of severity nor in a severity gradient. On the other hand, we think that these
data can support or reinforce the assertions made in this part of the discussion. These are
just two sentences with which we want to strengthen the idea that the persistence of an
altered pH and microbial profiles almost a decade after the fire may be in part a reflection
of the moderate to high severity of the fire in the sampled area. Something that has been
observed in shorter term works by Adkins et al. and Whitman et al. (Refs: 10, 11, 15, 16).
L479-486, closing paragraph a little weak, future monitoring is always important.
Reply: We believe that if the populations in both conditions (burned and control) were
similar, future monitoring would not be necessary. Therefore, we consider that the
statement made in this paragraph is well justified and makes a lot of sense in view of the
results presented. Anyway, future monitoring is always important, for example to know
when there is a total recovery with similar microbial communities in both forests
situations. In this sense a new sentence is added at the end of the conclusion section.

Reviewer 2 Report
The article verifies the changes in species and abundance of key microorganisms in soil ecological restoration after fire by examining the effects of fire on the structure of forest soil microbial communities over a long period of time. The findings of the article are very interesting and informative. However, there are still some problems that need to be solved.
1. The basic overview of the study site is discussed in Introduction, but the relevant cutting-edge introduction on the effects of fire on soil and ecosystem is missing.
2. Materials and Methods does not give a brief overview of the test site, what is the sampling method of the soil in the experiment, and what are the physical and chemical properties of the original soil. What is the age of the oak forest investigated in the experiment? It is necessary to add the relevant part of data.
3. What is the basis for choosing 9, 6 and 3 years for the experiment?
4. What is the correlation between the changes in microbial community structure and abundance analyzed in the Discussion section, but what is the correlation between the changes in soil physicochemical characteristics? A correlation analysis is missing.
5. The results of the post-fire vegetation restoration and strong changes in rainfall on the microbial community in the Results section of the article 3.6 are not covered in the Discussion.
6. The description of the research innovation points in the article is not too prominent and needs to be specified in the introduction.
Author Response
Reviewer 2
Comments and Suggestions for Authors
The article verifies the changes in species and abundance of key microorganisms in soil
ecological restoration after fire by examining the effects of fire on the structure of forest soil
microbial communities over a long period of time. The findings of the article are very interesting
and informative. However, there are still some problems that need to be solved.
1. The basic overview of the study site is discussed in Introduction, but the relevant cutting-edge
introduction on the effects of fire on soil and ecosystem is missing.
Reply: According to this reviewer comment we have added this paragraph to the
introduction section “To the best of our knowledge when a fire occurs, it induces a
reduction in diversity and an alteration in the composition of microbial communities, as
many of them are unable to withstand this disturbance in the short and intermediate term
[10,11]. Such changes are evident both in terms of the composition and functionality of
the community, with an increased abundance of copiotrophic microorganisms that
intensifies with the fire severity [15]. Furthermore, during these post-fire periods, an
enrichment in the metagenome of individuals exhibiting pyrophilic traits is observed. In
other words, as previously mentioned, these microorganisms possess accelerated growth
rates, but also enhanced heat tolerance, and the capability to process pyrogenic
compounds [12]. Consequently, in the short to intermediate term, the microbial community
displays greater resilience to post-fire conditions.”.
2. Materials and Methods does not give a brief overview of the test site, what is the sampling
method of the soil in the experiment, and what are the physical and chemical properties of the
original soil. What is the age of the oak forest investigated in the experiment? It is necessary to
add the relevant part of data.
Reply: That information is in section 2.1 from Materials and Methods, here is a subtraction
of this information “The two sampling sites were located on a steep slope facing south.
Within each study site, three sampling plots were randomly selected along 1.0 km
transects. The rhizosphere of three trees was collected per plot (2 sites x 3 plots/site x3
trees/plot = 18 trees), each tree with a diameter of at least 15 cm at breast height and
separated by at least 5 m from one another. The upper layer (first 5 cm) of soil was removed
and rhizosphere soil samples were collected (5 to 20-cm depth) following the main roots
of each plant until finding non-suberified roots, where we manually collected the soil firmly
attached to the roots.”. Physicochemical properties are in Table 2. There is no previous
data because the area was not studied until the fire occurred. We added information about
the age of the oak forest according to the reviewer comment.
3. What is the basis for choosing 9, 6 and 3 years for the experiment?
Reply: We were unable to sample before 3 years due to funding and logistical problems.
Thus, we decided to do long-term monitoring with this time span within each subsequent
sampling. Moreover, previous studies reach near 10 years after wildfire, so this study was
in the same framework but with NGS techniques and not fingerprinting, that it is to say we
can know what microorganisms are present and how is its evolution along time.
4. What is the correlation between the changes in microbial community structure and abundance
analyzed in the Discussion section, but what is the correlation between the changes in soil
physicochemical characteristics? A correlation analysis is missing.
Reply: We have seen that these practices are similar and common in other works cited
here (References 15 and 16).
5. The results of the post-fire vegetation restoration and strong changes in rainfall on the microbial
community in the Results section of the article 3.6 are not covered in the Discussion.
Reply: We disagree. In fact, this is mentioned in the 3rd paragraph of Discussion “With
respect to the taxonomic profiles and their evolution, a decrease of an abundant phylum,
Actinobacteria, is observed over time in both conditions (BOF and UOF). This pattern can
be related to a change in the rainfall regime, as the first sampling time was the driest,
accounting for both the beginning of the hydrological cycle up to the sampling date of
each year and the 4 months prior to each sampling event. In fact, a higher proportion of
Actinobacteria is a sign of more xeric conditions in a Mediterranean climate, as many
species of this phylum are known to produce spores and/or possess forms of resistance
that allow them to withstand hostile conditions such as drought or water limitation [9,48].
Nevertheless, the enrichment of this phylum in the burned area compared to the control is
still evident in the last sampling time (9 yaf).” We would not elaborate on this part of the
discussion in view of reviewer 1's comments in this regard.
6. The description of the research innovation points in the article is not too prominent and needs
to be specified in the introduction.
Reply: According to this reviewer comment, we added the following sentence at the last
paragraph of introduction “Given the limited number of long-term monitoring studies
conducted on post-fire forests, and considering that the existing studies are reliant on prehigh-
throughput sequencing methods, it is imperative to undertake research employing
contemporary techniques. Such studies would enable us to acquire a more
comprehensive understanding of the modifications experienced by rhizosphere microbial
communities inhabiting these environments.”.

Round 2
Reviewer 1 Report
Line numbers refer to track change version of the manuscript.
Introduction and conclusion much better. I couldn’t find the new supplementary Figure 1 (map of study area).
Holm oak forests or woodlands? Authors refer to them inconsistently across the manuscript (including abstract and introduction).
L61-68, need supporting references for this new text
L119, replace ‘suffered’
L123, locations, not location
Table 2, calculation of error terms for pH values is still incorrect – see original comment
acceptable
Author Response
- Introduction and conclusion much better. I couldn’t find the new supplementary Figure 1 (map of study area).
Reply: We uploaded a Zip file with the tracked and reviewed versions of the manuscript together with Figure S1 (map) and Figure S2 (previously Figure S1). Anyway, we have prepared a single pdf file with both figures.
- Holm oak forests or woodlands? Authors refer to them inconsistently across the manuscript (including abstract and introduction).
Reply: The term “woodlands” is only mentioned in Abstract and we consider it necessary to make sense of the sentence without generating too much redundancy. The sentence in question is "Yet knowing the time scale of the effects is indispensable to aiding post-fire recovery in vulnerable woodlands like holm oak forests subjected to Mediterranean climate, such as those found in protected areas like the Sierra Nevada National and Natural Park in southeastern Spain".
- L61-68, need supporting references for this new text
Reply: According to reviewer’s comment references 1, 3 and 6 were added to line 63.
- L119, replace ‘suffered’
Reply: changed “suffered a wildfire” by “where a wildfire occurred”
- L123, locations, not location
Reply: changed according to reviewer comment
- Table 2, calculation of error terms for pH values is still incorrect – see original comment
Reply: For pH, the company gave us a value for each sample. In Table 2 we show the mean and standard deviation of the 3 samples/replicates for each condition (control and burned).
